# Omics Strategies in Current Advancements of Infectious Fish Disease Management

**DOI:** 10.3390/biology10111086

**Published:** 2021-10-22

**Authors:** Maya Erna Natnan, Yosmetha Mayalvanan, Fahmeeda Mohd Jazamuddin, Wan Mohd Aizat, Chen-Fei Low, Hoe-Han Goh, Kamalrul Azlan Azizan, Hamidun Bunawan, Syarul Nataqain Baharum

**Affiliations:** Institute of Systems Biology (INBIOSIS), Universiti Kebangsaan Malaysia, UKM, Bangi 43600, Selangor, Malaysia; ernamay27@gmail.com (M.E.N.); yoshie.mayalvanan@gmail.com (Y.M.); fahmeedamojan@gmail.com (F.M.J.); wma@ukm.edu.my (W.M.A.); low@ukm.edu.my (C.-F.L.); gohhh@ukm.edu.my (H.-H.G.); kamalrulazlan@ukm.edu.my (K.A.A.); hamidun.bunawan@ukm.edu.my (H.B.)

**Keywords:** multi-omics, fish disease, aquaculture, metabolomics, proteomics, biomarker

## Abstract

**Simple Summary:**

The increasing demand for fish products has caused disease-related problems due to intense fish practices in fish farms. The fish diseases infection caused by bacteria, viruses, and parasites could lead to high fish mortality that affected the aquaculture industry. Recently, a systematic strategy to overcome fish disease problems using multi-omics platforms has been used to provide a better understanding of how to improve the resistance of fish to pathogen infection. In this review, we highlight the current multi-omics strategies such as transcriptomics, proteomics, and metabolomics to provide information regarding their molecular mechanisms of action, subsequently important in discovering potential biomarkers for various infectious fish diseases in the aquaculture system.

**Abstract:**

Aquaculture is an important industry globally as it remains one of the significant alternatives of animal protein source supplies for humankind. Yet, the progression of this industry is being dampened by the increasing rate of fish mortality, mainly the outbreak of infectious diseases. Consequently, the regress in aquaculture ultimately results in the economy of multiple countries being affected due to the decline of product yields and marketability. By 2025, aquaculture is expected to contribute approximately 57% of fish consumption worldwide. Without a strategic approach to curb infectious diseases, the increasing demands of the aquaculture industry may not be sustainable and hence contributing to the over-fishing of wild fish. Recently, a new holistic approach that utilizes multi-omics platforms including transcriptomics, proteomics, and metabolomics is unraveling the intricate molecular mechanisms of host-pathogen interaction. This approach aims to provide a better understanding of how to improve the resistance of host species. However, no comprehensive review has been published on multi-omics strategies in deciphering fish disease etiology and molecular regulation. Most publications have only covered particular omics and no constructive reviews on various omics findings across fish species, particularly on their immune systems, have been described elsewhere. Our previous publication reviewed the integration of omics application for understanding the mechanism of fish immune response due to microbial infection. Hence, this review provides a thorough compilation of current advancements in omics strategies for fish disease management in the aquaculture industry. The discovery of biomarkers in various fish diseases and their potential advancement to complement the recent progress in combatting fish disease is also discussed in this review.

## 1. Introduction

The exponential growth of the human population has markedly increased the global demand for food, particularly protein sources from an animal such as fish. However, continuous harvesting of wild fish has led to overexploitation of the wild stock and resulted in a great loss of fish species [1]. Aqua culturing was introduced to prevent overfishing from reducing the depleting wild fish stock. Fishes are bred in a controlled environment where they are subjected to routine feeding and are closely monitored to ensure longevity [2]. The aquaculture industry is driven by an ever-increasing demand for fish by most consumers from developed countries [3]. Proper management and established breeding technology are essential for the fish industry to fulfill the demand [4].

One of the most challenging challenges in sustainable aquaculture is managing and controlling infectious diseases [5]. Fish exposure to pathogens is even more severe and direct than non-aquatic organisms, considering that there are approximately one million bacteria and ten million viruses per milliliter of seawater. Fish are exposed to pathogens immediately after hatching and are continually affected during their mouth and gut opening stages, especially during the onset of their feeding [6]. They are also exposed to unknown pathogens when they migrate from freshwater to saltwater and during climatic change, including the non-migratory species [7]. Furthermore, fish disease outbreaks can be driven by extreme stress from the aquaculture environment and management procedures [8]. Despite monitoring the health of the fish stock due to the development and advancement of aquaculture, a continuous supply of fish products could not be supplied globally, primarily due to disease outbreaks [9].

Fish such as groupers with a high market value and are reared mainly in fish farms are exceptionally susceptible to infection [10]. The main pathogen that infects marine species is bacteria, which comprises 54.9% of the total infection followed by virus infection (22.6%), parasites (19.4%), and fungi (3.1%) [8,11,12]. The outbreak of these infections’ diseases in the large-scale fish farms will cost the farmers their revenue and offsets their business. While more money is churned out to rectify the disease, the turnover rate of the farm is being affected due to the decreased production of fish. However, as in all vertebrates, fish have cellular and humoral immune responses and organs as a defense against various pathogenic and non-pathogenic attacks [13,14]. Unlike mammals, fish are more dependent on the non-specific, innate defense system [15]. They are equipped with natural barriers that act as protective mechanisms: their skin and scales and the lytic proteins present in the mucus and sera [16]. While the innate defense system responds faster than the adaptive immunity of the species towards any foreign attack [17], the adaptive response of fish is essential for long-term immunological memory despite being implemented with a slight delay [18].

In this review, various fish pathogens and how fish respond to such attacks are explained. Apart from that, the current omics technologies used to control this problem, the multi-omics approaches were also briefly discussed, and how these emerging technologies are relevantly applied towards combatting various fish diseases. The review also focuses on biomarker discovery and the potential advancement it foreshadows in complementing the omics approach.

## 2. Fish Pathogens

Three main pathogens inflict the most damage to the aquaculture industry by contributing to mass cases of fish mortality: bacteria, viruses, and parasites. *Vibrio* hemorrhagic septicemia or vibriosis has a widespread infection in cultured groupers in Brunei Darussalam, Malaysia, Taiwan, Indonesia, Kuwait, Thailand, Singapore, and the Philippines [19]. The causative pathogens of vibriosis are from Gram-negative bacteria of *Vibrio* species, which include *Vibrio parahaemolyticus*, *Vibrio alginolyticus*, *Vibrio vulnificus*, *Vibrio carchariae*, *Vibrio anguillarum*, *Vibrio ordalii*, *Vibrio harveyi*, *Vibrio mimicus*, and many more from the family of Vibrionaceae [16,20,21]. The lack of study on vibriosis caused by other uncommon *Vibrio* species could impede further progression and development to prevent and control vibriosis measures in aquaculture [22]. This disease has seriously affected the aquaculture industry leading to significant loss of various species of cage-cultured fishes and shellfish production worldwide [23,24]. It was reported that vibriosis has particularly increased the rate of mortality among mariculture groupers [25]. In Asia, several major vibriosis outbreaks were reported such as in Vietnam, where farmed barramundi was infected with *V. harveyi* resulting in mortality up to 40% [26]. In a local farm of Sabah, Malaysia, a vibriosis outbreak occurred in Asian seabass that was caused by *V. harveyi* [27]. Other cases of vibriosis also occurred to other fish species such as grouper, *Epinephelus awoara* [28], sole, *Solea tauvina* [29], redbanded seabream [30], gilthead seabream, and European seabass [31]. The initial symptoms of the disease are lethargy, loss of appetite, darkening of the fish coloration, losing equilibrium, and exhibiting abnormal swimming behavior. The presence of ulcers on the skin, fin, and tails along with septicemia hemorrhage are also clinical signs and symptoms of vibriosis [12,22,32]. While vibriosis is one of the most common bacterial diseases to cause fish mortality, other bacterial diseases also infect a wide range of fish species.

A common bacterial infection is the furunculosis infection from *Aeromonas salmonicida*, a Gram-negative, non-motile rod bacterium that mainly affects salmonid species, trout, charr, and grayling. Stressors often trigger outbreaks of furunculosis caused by poor water quality, overcrowding, and rapid temperature changes [33]. Most pathological findings confirmed that the common signs of furunculosis are exophthalmia, skin hemorrhages, lethargy, darkened skin, abnormal swimming, ulcers as well as muscle hemorrhages in the liver, and necrosis lesions that developed from furuncles and boil on the skin [34,35]. From previous studies, several species from both salmonid species; Atlantic salmon (*Salmo salar*) and rainbow trout (*Oncorhynchus mykiss*) and non-salmonid species; turbot (*Scophthalmus maximus*) and halibut (*Hippoglosus hippoglosus*) had been studied for furunculosis [36,37,38,39]. In Atlantic salmon, *S. salar* infected by *A. salmonicida*, hemorrhagic septicemia caused high mortality rates, especially in juvenile and adult fish. It was known that hemorrhagic septicemia is often fatal almost immediately within two- or three-days exposure [38,40]. Meanwhile, in rainbow trout, the vaccination of subunit salmonicida strain A449 showed lower mortalities after three weeks of *A. salmonicida* post-infection [39]. Similarly, in turbot, furunculosis was treated with administering a suitable dosage of bacteriostatic antibiotic known as florfenicol [36].

Another bacterial infection is pasteurellosis that causes by the halophilic bacterium known as Photobacterium damselae subsp. piscicida. The bacterium is a Gram-negative bacterium that affecting a wide range of fish species such as seabass, cobia, yellowtail, seabream, and others [41]. The disease is commonly found in Japan, Europe, America, and Mediterranean countries [42]. It was reported that the disease usually occurs if the water temperature is over 20 °C [43]. The presence of this pathogen can be identified by the whitish tubercles or granulomas consisting of bacterial accumulation in the internal organs [42] The pasteurellosis outbreak can cause a high mortality rate that is responsible for high economic losses. 

Apart from bacterial infections, viruses also play a significant role in increasing the mortality rate of farmed fishes. Viruses make up most of the genetic diversity as they are the most abundant lifeforms found in the sea [44]. Their widespread presence is enough to prove that virus infections are unavoidable. The viral disease that marine fish species are especially susceptible to is viral nervous necrosis (VNN), which is also known as viral encephalopathy and retinopathy (VER) [45]. VNN is known to have infected a multitude of fish species worldwide. These include mostly grouper species that are especially susceptible to VNN infection, such as *Epinephelus coioide*, and *Epinephelus fuscoguttatus* [46,47]. Apart from groupers, other marine species such as *Anguilla* (Anguillidae), *Gadus morhua* (Gadidae), and *Umbriana cirrosa* (Sciaenidae) are a few examples amongst various other species that are susceptible to VNN. Due to its wide geographical and species distribution, the impact on the economy caused by VNN on the marine aquaculture industry is seriously damaging [48]. The causative agent of VNN or VER is widely known as the *Piscine nodavirus* of the genus Betanodavirus [49,50]. This Betanodavirus genus is known to have a high resistance in the aquatic environment that enables them to survive at low temperatures, and when an organism is infected, it is considered to be a virus reservoir that infects other organisms in its vicinity [51]. The clinical signs of VNN include lesions in the brain and retina, which ultimately lead to lethargy, discoloration, sight problem, abnormal movement, and loss of appetite [48].

Similarly, fish lymphocystis disease (FLD) is a global chronic viral infection over a wide range of water temperatures [52]. This disease was reported in *Epinephelus bruneus*, *Epinephelus malabaricus*, and *Epinephelus chlorostigma* cultured in marine net cages in Guangdong, China, and among *E. fuscoguttatus* in Malaysia [53]. It is caused by the iridovirus, fish lymphocystis disease virus (FLDV) [54]. Infected fish develop unique hypertrophied lymphocystis cells on skin, fins, and/or mouth that resemble small pearl-like nodules occurring clusters or singularly. The apparent nodules are due to the infected tissues enlarging disproportionately. Infected fishes are faced with a decreased market value due to tumor-like nodules. These nodules also interfere with the feeding of the fish, which simultaneously inhibits the fish growth rate [55,56].

While viruses require a host to sustain themselves, parasites are free-living invertebrate organisms known to be opportunistic pathogens. These free-living parasites do not necessarily require a host to survive and reproduce. There are also obligate parasites that need hosts to survive and reproduce. Even though both free-living parasites and obligate parasites infest fish species, the infestation of obligate parasites accounts for the severe mortality rate in fish [57]. Parasitical infections are not unheard of in large-scale fish farms. Dinoflagellate *Amyloodinium ocellatum* that causes amyloodioniosis (velvet disease) is among the easily noticeable parasitic infection due to the formation of grey patches on the skin and gills of the host [58]. The disease has been reported in Malaysia and Indonesia, affecting the *Epinephelus* spp. and *Cromileptes altivelis* [53]. Another parasitic disease reported infecting cultured fishes in Asian countries is cryptocaryonosis [59]. This disease is notoriously known to hinder the sustainable development industry of large yellow croaker, *Larimichthys crocea* [60]. It is commonly known as the white spot disease due to whitish or greyish spots that are nests of *Cryptocaryon irritans* found on the body surface and gills of the infected fish [59]. While the presence of pathogens in marine aquaculture is not always damaging, we cannot deny that a solution must be found to counter the increasing mortality rate of various fish species, especially those that contribute to the economic development of countries. Hence, multiple efforts have been engaged to refrain from this increasing fish mortality problem. In the next section, some current advancements and technologies used to manage the diseases are detailed.

## 3. Current Advancements in Infectious Fish Disease Management 

Managing the fish disease is the most crucial task for the farmers to alleviate fish losses. This has proven to be a challenge as several pathogens are responsible for increasing the mortality of farmed fishes. Fish farmers have incorporated pesticides, chemotherapeutic agents, and antibiotics in dealing with pathogenic infection on a large scale [61,62]. While all these strategies have their pros, the long-term disadvantages are overwhelming the advantages. Despite the continuous implementation of multiple methods, no progress has been made in improving fish health. It is crucial first to understand the fundamental biological system involved when a fish encounters an infection. What type of pathogen has infected the fish? How does the infection occur? What drives the disease progression? How does the fish resist the infection? These complex questions are possible to be addressed by first studying and understanding the molecular regulatory networks [63]. Recent developments have seen many effective preventive methods to contain diseases that aim to enhance fish immunity. These include vaccination and natural products possessing immunostimulatory properties [64]. In this section of the article, a few widely used approaches are highlighted to give an overview of how farmed fishes’ infectious disease problem is currently being managed.

### 3.1. Active and Passive Immunization

Organisms defend themselves from infection through resistance mechanisms that are activated by a series of immune responses. In cases where the organism is already infected, tolerance towards the infection contributes to fish survival by enabling the host to mount extraneous antibodies that temporary induce their immune system known as passive immunity [65,66]. Meanwhile, vaccination or known as active immunity is one of the most prominent approaches that have been used for strengthening host immunity and defending fish from both existing and future infections. First-generation vaccines are vaccines that are composed of whole pathogens. Live vaccines or attenuated pathogens can stimulate both cellular and humoral immune responses and these two vaccines are currently applied in the aquaculture industry [67]. Types of vaccines, their route of administrations, and the corresponding pathogens are shown in Table 1. The table also covers some of the diseases from multiple pathogens infecting different types of host species. The use of vaccines gained a lot of success initially as some of the fish diseases were able to be contained; however, not all vaccines were able to deter the effects of viruses completely. While vaccination is the most powerful method against disease outbreaks, external factors such as high production costs and antigenic heterogeneity of microbial strains hinder the development of effective vaccine formulations [67,68].

Similarly, antibiotics are a prevention method used to treat bacterial infections [69]. Antibiotic use in aquaculture may have less than desirable impacts on the environment and human health. This would involve the development and transfer of resistance of one aquatic bacteria to fish pathogens and aquatic bacteria. Apart from that, the accumulation of residual antibiotics in aquaculture products and the possibilities in the ways it could affect microbial biodiversity are also a concern [70].

**Table 1 biology-10-01086-t001:** Types of vaccines administered to fish.

Pathogens	Antigen Gene Insert	Host	Route	References
**RNA Viruses**				
Spring viremia of carp virus (SVCV)	pEGFP-G	Common carp (*Cyprinus carpio*)	Immersion & intramuscular	[71]
Viral Hemorrhagic Septicemia virus (VHSV)	pcDNA3-vhsG (DK-3592b, genotype Ia & BC-99-292, genotype IVa)	Pacific Herring (*Clupea pallasii*)	Intramuscular	[72]
Infectious hematopoietic necrosis virus (IHNV)	Glycoprotein	Rainbow trout (*Oncorhynchus* *mykiss*)	Intramuscular	[73]
**DNA Virus**				
Channel catfish virus (CCV)	DNA vector expressing CCV ORF6	Channel catfish	Intramuscular	[74]
Iridovirus of Taiwan (TGIV)	TGIV major capsid protein (MCP)	Pearl gentian grouper	Immersion	[75]
Koi herpesvirus (KHV)	ORF25 (glycosylated protein)	Koi	Intramuscular	[76]
**Bacterial**				
*Vibrio alginolyticus*	Lipopolysaccharides, whole-cell bacterin	Silver sea bream (*Sparus sarba*)	Intramuscular, immersion & oral	[77]
*Vibrio anguillarum*	Outer membrane proteins (OmpK)	Flounder (*Paralichthys olivaceus*)	Intramuscular	[78]
*Vibrio harveyi*	TssJ antigen from T6SS of *V. harveyi*	Golden pompano	Intramuscular	[79]

### 3.2. Immunostimulants

Immunostimulants can be synthetic or natural, derived from a chemical, drug, or naturally occurring compounds from animals and plants. For example, it has been reported that immunostimulants such as β-glucans occurring from the cell wall of yeast (*Saccharomyces* spp.) and levamisole, a synthetic antihelmintic used in mammals could enhance the disease resistance and give a better growth rate of tilapia, *Oreochromis niloticus* infected with *A. hydrophila* [80]. Immunostimulants are known to promote fish growth and immunity by stimulating relevant components such as lysozyme activity, complement activity, phagocytic activity, bactericidal activity, and respiratory burst activity in their immune system against diseases occurring in the fish species [81]. Immunostimulants can be administered in the fish body in three ways: injection, immersion, and oral uptake. Based on a previous study, oral delivery has been reported to be a long-established method and when compared to intraperitoneal injection, which is highly effective and enables the immunostimulants to be quickly absorbed, it is costly affairs with relatively time-consuming and labor-intensive [82]. In addition, despite the slow rate of product absorption by oral delivery, it is the most suitable method for fish farming as it is a non-stressful method and can treat a large number of subjects with the minimum cost and effort [82,83]. Although immunostimulants can induce the immune response and simultaneously increase the aquatic animals’ survival, one of the limitations of immunostimulants are the side effects manifested in fish due to the use of hormones and chemicals, which reportedly contribute to potentially dangerous residues for consumers [68].

### 3.3. Other Strategies

The selection of parental broodstock based on phenotypes of the species rather than the genotypes is now a thing of the past as traditional breeding eventually leads to inbreeding and loss of fitness [84]. Advances in genomics and transcriptomics have allowed genotypic parental brood selection. For example, advances in genomics have enabled the creation of hybrid species. Hybrid groupers are ideally more resistant to infection as they possess a stronger innate immune system [85]. One of the successful hybrids is *E. fuscoguttatus* (female) x *Epinephelus lanceolatus* (male). In another study, the alternative to control and avoid parasitical infection is by using pesticides. Bangladeshi farmers tend to use pesticides excessively to prevent Argulosis infection caused by *Argulus* sp. or fish lice. Argulosis is an infectious parasite for marine fishes that causes severe damages to the body of the fish and eventually leads to the organism’s death [86]. There is also the usage of probiotics in disease prevention in aquaculture. Probiotics are the members of the healthy microbiota associated with the host. According to Dawood et al. [87], a probiotic should benefit the host apart from being harmless. Characteristics that make up potential probiotics are non-invasive, non-pathogenic, and modifiable to industrial processes to facilitate commercial production. There are several mechanisms through which probiotics may prevent bacterial diseases. This includes producing compounds that create a hostile environment for pathogens, competing for essential nutrients and adhesion sites, and lastly modulating the immune responses of the hosts [88,89,90].

A combination of various current omics technologies can provide multiple insights on how the outbreak of infectious diseases among cultured fish could be tackled by offering a thorough understanding of the problem being researched. The abundance of huge amounts of quantitative data and the development of computational methodologies are the major motivations behind the emergence of systems biology [91]. Figure 1 illustrates the applications of multiple-omics technologies in fish disease studies, which will be discussed in the next section.

## 4. Multi-Omics Perspective on Infectious Fish Diseases Studies

The use of high-throughput omics approaches is becoming a powerful multidisciplinary tool for life science research, including fish disease studies. Omics approaches to study DNA variations (genomics), gene expressions at mRNA level (transcriptomics), protein expression (proteomics), and metabolite concentration (metabolomics) are essential for investigating the interplay between fish immune systems and pathogen infections [92]. The basic aims of these omics are to provide a particular conception and understanding of complex biological systems as a whole [93,94], especially on the infected fish. Furthermore, these approaches allow identifying gene networks and molecular pathways associated with the fish immune response towards pathogens [8]. Extensive applications of the multi-omics approach in fish disease studies could suggest multiple solutions to combat the increasing mortality of reared fishes on a cellular level as the current methods used, such as antibiotics and vaccinations are not sustainable in the long term.

Multi-omics or integration of omics data analysis provides a better understanding of molecular mechanisms of the fish immune system, regarding gene expression/regulation, protein expression, and metabolite production. The high-throughput data sets from different levels of omics studies can be used as a powerful method for the development of vaccines and immunostimulants that can be used in preventing and controlling disease infection in fish. The use of advanced technology such as next-generation sequencing (NGS), can directly be identified and specifically detect the presence of infection before any clinical signs appear in the fish [95]. In proteomics study, a high-throughput technology has been often used to identify novel antigens for new vaccine development. A study by Pang et al. [96], had identified outer membrane protein expression known as dihydrolipoamide dehydrogenase (DLD), to act as a vaccine candidate for *Epinephelus coioides* against three *Vibrio* species (*V. harveyi, V. alginolyticus, and V. parahaemolyticus*). In another study, tilapias induced with exogenous l-leucine elevated the serine and proline to eliminate *Streptococcus iniae* infection suggesting that the two metabolites play a crucial role in pathogen elimination by the host [97]. Elucidation of the mechanisms by either genomic, proteomic, and metabolomics would benefit the idea of the development of a novel vaccine or immunostimulant method to boost fish immune response from bacterial invasions, which simultaneously reduce or solve the fish disease problems.

As this review aims to compile the vast literature on the multi-omics approaches including transcriptomics, proteomics, and metabolomics for advancements management in fish disease studies, we will first describe each of the omics applications individually to assess their relevance used in fish’s immune response studies and then walk the readers through the biomarkers found from these studies. Table 2 compiles the studies on fish disease using various omics approaches. As each of the omics approaches uses various methods to analyze the available data, the table also gathers the methods used in each of the studies.

### 4.1. Transcriptomics

The transcriptomic approach has been extensively applied in many fish disease studies such as in grass carp (*Ctenopharyngodon idellus*) [98], rainbow trout (*O. mykiss*) [99], tilapia (*O. niloticus*) [100], pacific cod (*Gadus microcephalus*) [101], orange-spotted grouper (*E. coioides*) [102], brown-marble grouper (*E. fuscoguttatus*) [103] and others (Table 2). These studies have enabled the discovery of many immune-related genes and their involvement in immune-specific pathways that can eventually guide disease diagnosis and prevention. Previous studies conducted by [102,104] have utilized the RNA-seq technology to determine the transcriptional responses of orange-spotted groupers, *E. coioides* to Singapore grouper iridovirus (SGIV), *V. alginolyticus,* and *V. harveyi* infection, respectively. These studies highlighted the discovery of genes with altered expression in response to the infection, such as complement component-related genes, hepcidin-like antimicrobial peptide precursor, lectin, interleukins, and interferon-related genes. These genes were previously reported to have a significant role in the fish immune system [102,104]. Hepcidin is well-known as one of the important antimicrobial peptides (AMPs) in fish. It is an essential mediator for the innate immune response as it is usually present at the fish mucosal barrier and plays a key role in the fish’s first line of defense [102]. Hepcidin also catalyzes direct broad-spectrum antimicrobial activities, thus functioning in antimicrobial immune responses and iron homeostasis [105]. Besides hepcidin, previous studies have shown the importance of lectins as the recognition molecules that facilitate innate and adaptive immune responses. Lectins are known for their ability to bind carbohydrates due to the presence of their carbohydrate-recognition domain (CRD), which enables them to function as pathogen-recognition receptors (PRRs) and recognize the carbohydrates of bacteria [106,107]. This eventually leads to the other immunological functions of lectins such as cell agglutination, antiviral activities, phagocytosis, encapsulation, and production of antimicrobial peptides [106,107,108]. Interleukins and interferons are also important to the fish immune response and these groups of cytokines are mainly involved in inflammatory responses and antiviral defense, respectively [109,110,111,112].

Transcriptomic analysis of Asian seabass, *Lates calcarifer* on post-infection by nervous necrosis virus (NNV) [113] revealed an increase in the expression of genes involved in chemokine signaling pathways such as CC chemokines and CXC chemokines that are similar to a study in the liver of large yellow croaker, *Larimichthys corcea* infected against *C. irritans* [114]. The chemokines family has been recognized for its importance in many biological processes including innate and adaptive responses. Functionally, chemokines are divided into inflammatory chemokines and homeostatic chemokines [115]. The inflammatory chemokines such as CCL1-5 and CXXL1-11 generally regulate neutrophil and activated T-cell migration. Most CC chemokines are crucial for inducing chemotaxis monocytes, T-cells, and macrophages, leading to the innate immune response [116]. Meanwhile, homeostatic chemokines such as CXCL12 are essential for the migration of antigen-presenting cells (APC) and lymphocytes to the lymph node where immune surveillance occurs. These chemokines are also important for the migration of T cells to the tissues with APC. These actions are important for effective adaptive immune responses [116,117].

Transcriptomics approach not only provides an insight into the response of the fish towards different types of infection at the transcriptome level but also helps in revealing the complexity of the host’s defense mechanism against infection [103] by the identification of potential key resistance genes that is important to improve the immunity of fish for disease resistance. However, while transcriptome profiling can be comprehensive, it may be subjected to post-transcriptional and post-translational modifications. As such, proteomics is needed to verify and validate the protein levels.

### 4.2. Proteomics

Several reports have described the applications of mass spectrometry-based (MS-based) proteomics in studying the fish proteome response such as in Japanese flounder (*Paralichthys olivaceus*) [118], common carp (*Cyprinus carpio*) [119], Atlantic salmon [120], and many more (Table 2). A comparative analysis of protein expression in the spleen of healthy and infectious spleen and kidney necrosis virus (ISKNV) infected mandarin fish (*Siniperca chuatsi*) revealed the up-regulated autophagy-related proteins including LC3 and P13Ks indicated the induction of CPB cells (Chinese perch brain cells) autophagy in the early-stage infection [121]. Meanwhile, the heat-shock protein 27 (HSP27) family was found to be up-regulated in response to grouper nervous necrosis virus (RGNNV) infection in the sea perch [122]. The up-regulation was similar to other fish-virus and bacterial studies [123,124], highlighting it as a therapeutic target for virus or bacteria-associated diseases. Furthermore, the proteomics approach of zebrafish skin infected with *Aeromonas hydrophila* revealed the increased expression of actin, myosin heavy chain, and glyceraldehyde 3-phosphate dehydrogenase (GAPDH) [125]. The expression of both actin and myosin in the skin indicated that these proteins are important for the skin response of zebrafish to *A. hydrophila* infection. A few studies have shown that actins play significant roles in resistance to bacterial or viral infections [126]. The presence of actin such as F-actin and other small GTPases Rho, Rac, and cell division cycle 42 were reported to be important in cell motility of fish as it aids in initiating the phagocytic processes for invasion of pathogens via rearrangement of the actin cytoskeleton [127,128]. Myosin protein is composed of two heavy chains and two light chains with ATPase activity and actin-binding sites. It provides contractile force through hydrolysis of ATPs and interaction with actin [129,130]. Myosin was suggested in a previous study to be activated during an immune response. It was suggested that the increased disease resistance in fish might be associated with the low expression of myosin. A low expression of myosin was proposed to inhibit muscle growth, which inhibits further pathogen infection [125]. Meanwhile, GAPDH was reported to have multiple roles in immune function, defense responses, antioxidant, and energy metabolism in fish [131].

Other immune-related proteins expressed due to *V. anguillarum* infection are mannan-binding lectin (MBL) and cyclophilins in the skin mucus of Atlantic cod, *Gadus morhua* [132]. Thus, these proteomics studies have shown the importance of an advanced high-throughput proteomic approach in disclosing the complication raised regarding fish response towards disease infection at the protein level. However, this alone will not provide sufficient information to wholly understand all the underlying mechanisms that play a role in responding to a pathogenic threat. Metabolomics study complements the two omics as mentioned earlier in this way as the slightest change in the environment will affect the metabolites produced by an organism. This particular advantage is utilized in the many metabolomics studies that are discussed below.

### 4.3. Metabolomics

Similar to the other omics, metabolomics has continued to grow rapidly and is considered a prominent tool that predicts and explains the complex phenotypes in a diverse biological system. It acts as a complementary platform and is often used in accordance with transcriptomics and proteomics since metabolomics is a downstream result of gene and protein expression. Metabolomics utilizes the high-throughput screening of various samples using instruments to assess any chemical risk. A study by Du et al. [97] reported on the effects of l-leucine-induced metabolome to eliminate *Streptococcus iniae* in tilapias, exploring the metabolomics approach’s potential in infectious diseases. Other studies on the metabolome changes during bacterial infection also have been conducted on several fish species such as tilapia [133], Atlantic salmon [40,134], crucian carp [135], and grouper [136]. A recent study by Low et al. [49], discussed the limitations and the challenges of metabolomics applications in fish disease studies while emphasizing the impacts of infectious fish diseases and the possibility of enhancing disease resistance in fish using metabolomics.

Activities of the cells are reflected by the metabolome changes that can identify complex biologically essential changes [135]. In exploring targeted compounds and pathways necessary for crucian carps defense against infection caused by *Edwardsiella tarda*, a gas chromatography–mass spectrometry-based (GC-MS-based) metabolomics approach was used in this research. The research has highlighted elevation in unsaturated fatty acid biosynthesis and decreased fructose and mannose metabolism. In contrast, the increase in palmitic acid and decrease in D-mannose were highlighted as the most crucial metabolic difference. These pathways and metabolites were suggested to be the central biomarkers in differentiating survivals from death in crucian carps infected by *E. tarda*. Their findings stressed the importance of metabolic strategy in studying and understanding fish responses to bacterial infections. Thus, targeted compounds and pathways identified can be a good reference for further analysis on host-pathogen interaction.

In another study [136], the high concentration of amino acid such as valine and leucine detected in the susceptible group of groupers infected with vibriosis is related to the activation of anti-bacterial infection pathways that involves valine, leucine, and isoleucine metabolism. Identification of these metabolites (valine and leucine) and their specific functions in inhibiting or escalating the immune response has enabled multiple other research to be conducted in order to improve the current problems in fish disease. The regulatory role of amino acids such as leucine, isoleucine, and valine are reported in various fish studies centered around identifying the dietary and regulatory effects of amino acids in fish and their impact on the immune system [40]. In a study conducted by Castro et al. [6] that aimed to observe the immune responses at an early stage in rainbow trout (*O. mykiss*) infected with viral hemorrhagic septicemia virus, it was found that the liver is an immunocompetent organ that has a vital role in the immune response. Furthermore, a study conducted by Lardon et al. [137] on common carps (*C. carpio*) found that the liver contained metabolites associated with the general energy pathway such as ATP/ADP, phosphocreatine, lactate, and amino acids metabolites such as valine, (iso)leucine, alanine, glutamate, glutamine, glycine, and aspartate were significantly decreased in fish that were induced with hypoxia compared to control fish. The decreased of these metabolites was due to oxygen limitation availability, which minimize the consumption of ATP/ADP and simultaneously decreased the accumulation of lactate to conserve their energy consumption [138]. As for amino acids, the decrease of the metabolites was due to the use of amino acids in regulating the growth and metabolism of the fish [139]. While hypoxia is not an infectious disease, the lack of oxygen in the fish system is seen to achieve a similar stress response in the fish when infected with pathogens, according to the results of a study conducted by Lardon et al. [140]. Since amino acids are vital to regulating key metabolic pathways related to maintenance and immune responses, identifying specific amino acids unique to certain biological pathways is essential to discover biologically active metabolites [136,137]. In another study, the effects of waterborne chlorpyrifos in freshwater carp, *Cyprinus carpio* had caused the increased biosynthesis of valine, leucine, and isoleucine for oxidative/ stress metabolism, while the increased of, alanine and lactate pattern suggests that the subsequent metabolic pathway was inhibited. An increase of pyruvate is found to inhibit the metabolic pathway leading to the acetyl-CoA formation that will impact the tricarboxylic acid cycle (TCA cycle) and in turn the energy metabolism of the fish. As a large part of the energy is allocated to maintain and activate the immune system, inhibition of the TCA cycle will undoubtedly cause a delay in the immune response of the infected fish [141].

The roles played by metabolites are again emphasized in a metabolomics analysis conducted by Liu et al. [40] to profile the metabolites in the kidneys of Atlantic salmon infected with *A. salmonicida*. It was found that the metabolites extracted from the infected samples have an altered profile compared to the control ones. Four pathways were reconstructed from the significantly altered metabolites (fumarate, alanine, valine, glycine, choline, glycerophosphorylcholine (GPC), and aspartate) that were obtained from the proton nuclear magnetic resonance (H-NMR) analysis. These metabolic pathways are the citrate cycle, glycolysis/gluconeogenesis, tryptophan metabolism, and the urea cycle. While alanine and glycine are found to be upregulated mid-experiment, valine shows the opposing result. While not directly affecting the innate immune system, these cycles play a role in producing the intermediates. This study by Liu et al. [40] suggests that alanine and glycine might play a role in protein synthesis, benefiting the viral genome synthesis.

A study by Nurdalila et al. [142] that involved metabolite fingerprinting of *E. fuscoguttatus* infected with *V. vulnificus* had fascinating findings where omega 9 (ω-9) was found to be a potential biomarker for vibriosis in brown marbled groupers. Regulation of immune cell function by short-chain fatty acids with the host immune response and pathogen resistance may be influenced by fatty acids that play a significant role as demonstrated by human and animal studies, during ex vivo and in vitro experiments [143]. This study was essentially based on the speculation that fatty acids induce changes in immune responses by initiating various processes such as altering membrane fluidity, lipid peroxide formation, eicosanoid production, and gene regulation. With the combination of these processes and their effects on the immune system, dietary fatty acids influence pathogen clearance. In another study, omega-9 (ω-9) fatty acids have been proven to boost the high-density lipoprotein cholesterol while reducing low-density lipoprotein [144]. Omega-9 fatty acids are categorized as polyunsaturated fatty acid (PUFA), which regulates prostaglandin synthesis and induces wound healing. This explains the relevance of ω-9 fatty acids being considered as a potential biomarker for vibriosis in brown-marbled groupers [145], which is also supported by a study on *E. fuscoguttatus* by Nurdalila et al. [142] that reported an increased concentration of ω-9 compared to omega-6 (ω-6).

Confirmation of biomarker identity is the major bottleneck in metabolomics investigations. What can be deduced from the various discussion regarding the roles of metabolites that are found to be notably upregulated or downregulated in the hosts and the roles they play inactivation of an immune response is that these amino acids collectively provide a chance to enhance further the existing approaches used to combat fish disease. For example, the fish feed can be improved in various ways where the metabolites that strengthen the immunity of fish can be incorporated into meals. Apart from that, knowledge about compounds that seemingly inhibit the immune response is also crucial for researchers to find a way around the inhibition so that the natural immune response of fish is not interrupted. There are many ways to apply a multi-omics approach in the aquaculture industry to enhance fish health and improve the current technologies involved in curbing the fish disease.

**Table 2 biology-10-01086-t002:** Compilation of studies on fish immune response using various omics approaches.

Fish Species	Pathogen	Organ/Tissue Samples	Method	Reference
	*Transcriptomics*			
Crucian carp *(Carassius auratus)*	*Aeromonas hydrophila* (bacteria)	Head kidney	Illumina Hiseq sequencer	[146]
Tilapia (*Oreochromis niloticus*)	*Streptococcus iniae* (bacteria)	Spleen	Illumina HiSeq 2000 instrument	[100]
Brown-marbled grouper (*Epinephelus fuscoguttatus*)	*Vibrio vulnificus* (bacteria)	Gill and whole-body tissue	Illumina HiSeq. 4000	[103]
Turbot (*Scophthalmus maximus*)	*Vibrio anguillarum* (bacteria)	Intestine	Illumina HiSeq 4000	[147]
Orange-spotted grouper (*Epinephelus coioides*)	*Vibrio alginolyticus* (bacteria)	Whole body tissue	Illumina HiSeq 2000	[102]
Rainbow trout (*Oncorhynchus mykiss*)	*Flavobacterium* (bacteria)	Spleen	Illumina TruSeq	[148]
Soiny mullet (*Liza haematocheila*)	*Streptococcus dysgalactiae* (bacteria)	Spleen	Illumina HiSeq 2000	[149]
Rainbow trout (*Oncorhynchus mykiss*)	*Ichthyophthirius multifiliis* (parasite)	Gill	Illumina HiSeq 2500	[99]
Large yellow croaker (*Larimichthys crocea*)	*Cryptocaryan irritans* (parasite)	Liver	Illumina HiSeq2000	[114]
Orange-spotted grouper (*Epinephelus coioides*)	*Cryptocaryon irritans* (parasite)	Skin	Illumina HiSeq 2500	[150]
Striped snakehead (*Channa striata*)	Red-spotted grouper nervous necrosis virus (RGNNV)	Striped snakehead fish cells (SSN-1)	Illumina HiSeq 2000	[151]
Atlantic salmon (*Salmo salar*)	Infectious salmon anemia virus (ISAV)	Spleen	Illumina MiSeq sequencer	[152]
Grass carp *(Ctenopharyngodon idellus)*	Grass carp reovirus (GCRV)	Kidney	Illumina NextSeq500	[98]
Asian seabass (*Lates calcarifer*)	Nervous necrosis virus (NNV)	Epithelial cells	Illumina HiSeq™ 2000	[113]
Koi (*Cyprinus carpio*)	Cyprinid herpesvirus 3 (CyHV3)	Spleen	Illumina HiSeq 2500	[153]
Pacific cod (*Gadus microcephalus*)	General/Not specified	Thymus and head kidney	Illumina HiSeq 2000 platform	[101]
	**Proteomics**			
Brown-marbled grouper (*Epinephelus fuscoguttatus*)	*Vibrio parahaemolyticus* (bacteria)	Blood	2D gel electrophoresis, Matrix-assisted laser desorption/ionization-time of flight-mass spectrometry (MALDI-TOF-MS/MS) analysis.	[16]
Rainbow trout (*Oncorhynchus mykiss*)	*Yersinia ruckeri* (bacteria)	Intestine	Micro Liquid chromatography coupled with electrospray ionization and quadrupole time of flight tandem- mass spectrometry (LC-ESI-qTOF-MS/MS) analysis.	[154]
Zebrafish (*Danio rerio*)	*Aeromonas hydrophila* (bacteria)	Skin	2D gel electrophoresis, MALDI-TOF-MS analysis, Liquid chromatography–mass spectrometry (LC-MS/MS) analysis.	[125]
Yellow catfish (*Pelteobagrus fulvidraco*)	*Edwardsiella ictalurid* (bacteria)	Skin mucus	LC-MS/MS analysis	[155]
Common carp (*Cyprinus Carpio*)	*Aeromonas hydrophila* (bacteria)	Intestine	LC-MS/MS analysis	[119]
Pufferfish (*Takifugu obscurus*)	*Aeromonas hydrophila* (bacteria)	Spleen	LC-MS/MS analysis	[156]
Japanese flounder (*Paralichthys olivaceus*)	*Edwardsiella tarda* (bacteria)	Liver	isobaric tags for relative and absolute quantification (iTRAQ) analysis, LC-MS/MS analysis.	[118]
Atlantic salmon (*Salmo salar*)	*Neoparamoeba* *Perurans* (parasite)	Gill	2D gel electrophoresis, LC-MS/MS analysis	[120]
Lumpsucker (*Cyclopterus lumpus*)	General/Not specified	Skin mucus	2D gel electrophoresis, LC-MS/MS analysis	[157]
Gilthead seabream (*Sparus aurata* L.)	General/Not specified	Skin mucus	2D gel electrophoresis, Peptide mass fingerprinting-mass spectrometry (PMF-MS/MS) analysis, LC-MS/MS analysis	[158]
	**Metabolomics**			
Atlantic salmon (*Salmo salar*)	*Aeromonas salmonicida* (bacteria)	Kidney	Nuclear magnetic resonance (H-NMR) analysis.	[40]
Brown-marbled grouper (*Epinephelus fuscoguttatus*)	*Vibrio vulnificus* (bacteria)	Caudal fin	Fourier-transform infrared spectroscopy (FTIR) analysis.	[4]
Zebrafish (*Danio rerio*)	*Vibrio alginolyticus* (bacteria)	Whole body tissue	Gas chromatography–mass spectrometry (GC-MS) analysis.	[159]
Brown marble grouper (*Epinephelus fuscoguttatus*)	*Vibrio vulnificus* (bacteria)	Muscle tissue	GC-MS analysis	[136,142]
Tilapia (*Oreochromis niloticus*)	*Streptococcus iniae* (bacteria)	Liver	GC-MS analysis	[133]
Tilapia (*Oreochromis niloticus*)	*Streptococcus iniae* (bacteria)	Liver	GC-MS analysis	[160]
Tilapia (*Oreochromis niloticus*)	*Streptococcus agalactiae* (bacteria)	Liver	ultraperformance liquid chromatography-tandem mass spectrometry (UPLC-MS) analysis.	[161]
Zebrafish (*Danio rerio*)	*Edwardsiella tarda* (bacteria)	Muscle tissue	GC-MS analysis	[162]
Japanese puffer (*Takifugu rubripes*)	*Cryptocaryon irritans* (parasite)	Blood serum	LC-MS analysis	[163]
Tiger puffer fish (*Takifugu rubripes*)	Myxosporea (parasite)	Blood serum	GC-MS analysis	[164]
Mandarin fish *(Siniperca chuatsi)*	Infectious spleen and kidney necrosis virus (ISKNV)	Chinese perch brain cell line	ultra-high-performance liquid chromatography-quadrupole time-of-flight mass spectrometry (UHPLC-Q-TOF/MS) analysis.	[165]
Grouper	Red-spotted grouper nervous necrosis virus (RGNNV)	Spleen	LC-MS analysis	[166]
Crucian carp blood *(Carassius auratus gibelio)*	*Cyprinid herpesvirus 2*	Blood	LC-MS analysis	[167]

## 5. Application of Multi-Omics for Identification of Biomarker

The emergent of these omics approaches widely used in fish disease studies facilitates understanding various fish disease mechanisms and discover different biomarkers of both the disease virulence and the host defense mechanism. These high-throughput technologies combined with bioinformatics can generate large amounts of data to speed up the identification of potential biomarkers for various diagnostic and therapeutic developments [168]. Genes, mRNAs, proteins, metabolites, and other molecules are types of potential molecular biomarkers that have been identified in numerous fish disease studies [92]. These biomarkers can outline and detect the differences between multiple factors that cause an organism to respond a certain way to an infection. Apart from that, biomarkers are used to detect the early stage of an infection and are used to interconnect the various mechanisms that relate to other factors with disease infection [169].

Table 3 shows a summary of biomarker discovery from the various studies using different omics approaches. A study made by Geng et al. [170] utilizes the genome-wide association study (GWAS) on catfish to identify the genes associated with columnaris resistance within quantitative trait locus (QTL). It was found that five genes of pik3r3b, cyld-like, adcyap1r1, adcyap1r1-like, and mast2 of linkage group 7 were significantly associated with columnaris resistance and known to have functions in immunity. These candidate genes may be arranged as functional hubs in the PI3K signal transduction pathway, suggesting their importance in the columnaris resistance trait. The detected and identified resistant genes from this study can be used as the targeted potential biomarkers responsible for the columnar-resistant trait catfish.

In Liu et al. [171] proteomic study, advancement of orbitrap coupled with ultra-high-performance liquid-chromatography/mass spectrometry (UPLC/MS) technology has eased the proteomic analysis in discovering a few immune-related proteins as the potential biomarkers to be used in the future for detection and prevention of disease infection in fish. Comparative proteomic analysis on the blood of Japanese puffer (*Takifugu rubripes*) upon infection by *C. irritans* identified few immune-relevant proteins that exhibited significant differential expressions such as integrin beta-1-like isoform X2, H2A.V, glucokinase-like, H4, histone H1-like, histone H2AX-like, histone H2B ½-like and myosin-9 isoform X1. All these identified proteins were suggested to be considered as the potential biomarkers in studies involving *C. irritans* infection.

Peng et al. [172] performed a metabolomic study also identified 11 metabolites as potential survival biomarkers. However, glucose was the most significant metabolite expressed in surviving and dying juvenile tilapias infected by *E. tarda*. The usage of the metabolomics approach in biomarker discovery is supported by the assumption that metabolites are important players in illuminating key biochemical pathways and biological systems caused by disease infections [173,174]. However, the current strategy in metabolomics to find single biomarkers for a disease is hampered by the highly fluctuating dynamic of metabolites where entire metabolic pathways change instead of one metabolite [175]. Even in the before mentioned experiment conducted by Liu et al. [40], they proposed that maybe one or more biomarkers play a role in disease activation and immune system regulation. Conventionally, biomarker development involves a discovery phase, typically conducted by mass spectrometry (MS) and followed by validation. While this approach has been tried and tested for the development of single biomarkers, the current drive is towards larger panels of multiplexed biomarkers, thus rendering the process of developing single biomarkers inefficient and not to mention costly [176].

## 6. Conclusions

Omics approaches have significantly been utilized to study pathogen-host mechanisms and interaction as well as vaccine development. The omics approach is also being employed to improve the fish diet by incorporating the required supplements to promote and increase immune response. Despite a few limitations, these approaches are generally reliable and widely used for many disease-related studies apart from fish diseases. A combination of these omics has the potential to solve problems regarding fish disease and promote fish health. Applications of transcriptomics, proteomics, and metabolomics, in particular, are impacting the increasing rate of mortality and stabilizing the fish export economy. The integration of different omics is expected to give a better understanding of fish immune mechanisms during disease infection, where different layers of omics data generated can be used to reflect the characteristic of an organism at different biomolecule levels. With the reliable database and easy outreach, the emerging field of integrative data analysis using of bioinformatics approach will produce a reliable and accurate diagnosis toward fish disease biomarkers. Though there is a lot of room for improvements as there remain various pathogens and fish species to be covered, all the major species of fish used in import and export are being studied currently. These studies are generating progressively exciting findings that are sparking off many further studies. In a nutshell, this review highlights the various efforts in combating fish disease through the applications of multi-omics technologies and discusses the mainstream approaches applied. This review also manages to steer the readers towards appreciating multi-omics, a relatively new technology currently progressing towards biomarker discovery to aid disease management in aquaculture.

## Figures and Tables

**Figure 1 biology-10-01086-f001:**
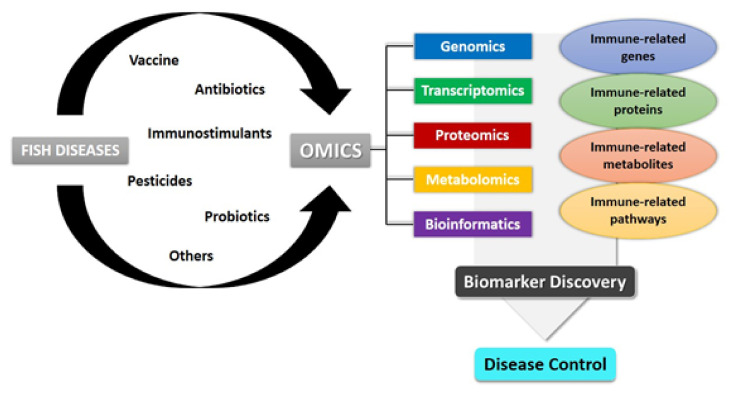
Applications of omics in the current technologies available for fish diseases.

**Table 3 biology-10-01086-t003:** Biomarker discovery using omics approaches.

Omics Approach	Fish/Infection	Organ/Tissue Samples	Technique	Potential Biomarkers	Study
Genomics	Channel catfish (*Ictalurus punctatus*) Columnaris disease (*Flavobacterium columnare*)	Blood	Affymetrix Axiom genotyping array technology	Genes: *pik3r3b* *cyld-like* *adcyap1r1* *adcyap1r1-like* *mast2*	[170]
Proteomics	Japanese puffer (*Takifugu rubripes*) (*Cryptocaryon irritans)*	Blood serum	Orbitrap coupled to UPLC/MS analysis	Proteins: integrin beta-1-like isoform X2, H2A.V, glucokinase-like, H4, histone H1-like, histone H2AX-like, histone H2B ½-like myosin-9 isoform X1	[171]
Metabolomics	Juvenille tilapias Edwardsiellosis (*Edwardsiella tarda*)	Liver	GC-MS analysis	Metabolite: glucose	[172]
Metabolomics	Atlantic salmon Furunculosis (*Aeromonas salmonisida*)	Kidney	H-NMR analysis	Metabolite: alanine glycine valine choline glycine Betaine	[40]
Metabolomics	Tilapias *(Streptococcus iniae)*	Liver	GC-MS analysis	Metabolite: N-acetylglucosamine	[133]
Metabolomics	Brown-marble grouper (*Epinephelus fuscogutttus*) Vibriosis (*Vibrio vulnificus*)	Muscle tissue	GC-MS analysis	Metabolite: leucine valine 8,11-eicosadienoic acid 6,9-octadecenoic	[136]

## Data Availability

No new data were created or analyzed in this study. Data sharing is not applicable to this article.

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
