# Peer review of "Omics Strategies in Current Advancements of Infectious Fish Disease Management"

_biology, 2021, doi:10.3390/biology10111086_

Round 1
Reviewer 1 Report
The review describes the applications of OMICs approaches to diagnose fish diseases.
There are some points that should be revised:
1- Section 3-1: No mention of antibodies, or passive immunity.
It would be fine if you replace vaccines and antibodies with "active and passive immunizations".
2- Table 3: Make the study column a right-sided column.
3- In tables 2 and 3, it is necessary to know the sampling (organ or water or which tissue)? The problem with diagnosis is the representative and correct samples!!
4- All abbreviations in the tables and text should be clarified and spelled.
5- I am conservative about using only metabolomics to detect the diseases, metabolomics can indicate the progress of disease or its cure, and its individual use as a diagnostic tool is unreliable.
6- In the conclusion, please show your point of view about the "combined" use of different omics to reach a reliable and accurate diagnosis.
Author Response
Reviewer 1
The review describes the applications of OMICs approaches to diagnose fish diseases.
There are some points that should be revised:
1- Section 3-1: No mention of antibodies, or passive immunity.
It would be fine if you replace vaccines and antibodies with "active and passive immunizations".
Response 1: Rephase the sub-section as suggested. Please refer to the manuscript page 5, line 205.
2- Table 3: Make the study column a right-sided column.
Response 2: The study column was changed to right-sided column. Please refer the revised manuscript Table 3, page 18.
3- In tables 2 and 3, it is necessary to know the sampling (organ or water or which tissue)? The problem with diagnosis is the representative and correct samples!!
Response 3: Added new column for ‘organ/tissue sample’ used for analysis in table 2 and 3. Please refer the revised manuscript Table 2, page 12 and Table 3, page 18.
4- All abbreviations in the tables and text should be clarified and spelled.
Response 4: All abbreviations were clarified and spelled in the text and table.
5- I am conservative about using only metabolomics to detect the diseases, metabolomics can indicate the progress of disease or its cure, and its individual use as a diagnostic tool is unreliable.
Response 5: Thank you for your view and concern of using individual metabolomic approach as a diagnostic tool in fish disease problem. However, in recent years, the advancement uses of high-resolution, accurate-mass MS (mass spectrometry) instrument such as the orbitrap, have emergence of not only accurate and sensitive metabolomic results but also provides rapid analytical tool for metabolite quantitation and identification. The advancement of this metabolomic platform have drawn new insight for biomarker discovery, which targets metabolic reaction and can be used as a diagnostic tool for disease identification in fish. Yes, endeavours and other reliable tool might need to employ along the metabolomic approach to help researchers in gaining more insightful information for better understanding of biological system and molecular processes in an organism. Nevertheless, metabolome provides the closest link to the phenotype of an organism and other advantages of MS analysis are the number of metabolites appears to be considerably lower than the number of gene and protein expression in genomic and proteomic study, which easier for researchers to analyse the data.
6- In the conclusion, please show your point of view about the "combined" use of different omics to reach a reliable and accurate diagnosis.
Response 6: We added statements on this point of view. Please refer to the conclusion section in the manuscript page 20, line 643-648.
Reviewer 2 Report
The review article by Natnan et al., discusses the use of the multi-omics approach for disease management in aquaculture. This review article will be a useful resource for researchers in the fisheries and aquaculture field. The article summarized various bacteria, viral, and parasite-associated infections in fisheries and their impact on fish health and the economy of fish industries. The authors provided sufficient materials and references to explain the impact of omics studies in understanding the host-pathogen interactions between different fish species and pathogens. Using the two tables’ authors summarized the different omics platforms used widely for studying the immune response and the biomarkers developed using these studies that can be used for developing interventions. Despite the great work by the authors in compiling in the review article, there are several typos and mistakes in sentence framing that can be improved to make this review article easier for the readers. I mentioned a few of my concerns below. I request the authors to thoroughly revise the manuscript for typos and sentence errors. I am happy to recommend the acceptance of this review article.
Minor corrections
Line 296: remove in the previous study
Lines 312-313: remove "of cytokines that are"
Line 314- That are similar – correct the typo
Line 340: induction of
Lines 384-386: Please rephrase this sentence into a simple sentence. It is confusing for now
Lines 387-390: Break this sentence into two and cite the reference appropriately
Line 394: death in crucian carps- correct the typo
Lines 398-400: is this part of the same study cited previously in the paragraph? If not include the citation. Or move this sentence to an appropriate place.
Line 401- these metabolites – what are the metabolites?
Line 412: significantly different? Are they increased or decreased? What are the implications of these changes on the health and survival of fish?
Line 416: in regulation or to regulate – correct the typo
Lines 418-421: these sentences do not correlate and do not provide any meaningful information- authors should correct this.
Line 501- also identified 11 metabolites – correct this
Author Response
Reviewer 2
The review article by Natnan et al., discusses the use of the multi-omics approach for disease management in aquaculture. This review article will be a useful resource for researchers in the fisheries and aquaculture field. The article summarized various bacteria, viral, and parasite-associated infections in fisheries and their impact on fish health and the economy of fish industries. The authors provided sufficient materials and references to explain the impact of omics studies in understanding the host-pathogen interactions between different fish species and pathogens. Using the two tables’ authors summarized the different omics platforms used widely for studying the immune response and the biomarkers developed using these studies that can be used for developing interventions. Despite the great work by the authors in compiling in the review article, there are several typos and mistakes in sentence framing that can be improved to make this review article easier for the readers. I mentioned a few of my concerns below. I request the authors to thoroughly revise the manuscript for typos and sentence errors. I am happy to recommend the acceptance of this review article.
Minor corrections
Line 296: remove in the previous study
Response 1: The statement was removed as suggested. Please refer page 8, line 335.
Lines 312-313: remove "of cytokines that are"
Response 2: The statement was removed as suggested. Please refer page 8, line 350
Line 314- That are similar – correct the typo
Response 3: Added ‘are’ in the sentence. Please refer page 8, line 351.
Line 340: induction of
Response 4: Mistake has been corrected, ‘inducing’ to ‘induction’. Please refer to page 9, line 380.
Lines 384-386: Please rephrase this sentence into a simple sentence. It is confusing for now
Response 5: We are sorry for the confusing. We have rephase the sentence to ‘Other studies on the metabolome changes during bacterial infection also have been conducted on several fish species such as tilapias [132], Atlantic salmon [40,133] and groupers [135].’ Please refer to the manuscript page 10, line 422-425.
Lines 387-390: Break this sentence into two and cite the reference appropriately
Response 6: The sentence is split into two sentences and rephase for better understanding. ‘Activities of the cells are reflected by the metabolome changes that capable of identifying complex biologically essential changes [134]. In exploring targeted compounds and pathways necessary for crucian carps defence against infection caused by Edwardsiella tarda, GC-MS-based metabolomics approach was used in this research.’ Please refer to the manuscript page 10 line 429-433.
Line 394: death in crucian carps- correct the typo
Response 7: The typo has been corrected. Please refer to the manuscript page 10, line 437.
Lines 398-400: is this part of the same study cited previously in the paragraph? If not include the citation. Or move this sentence to an appropriate place.
Response 8: Is not part of the same study. Citation has been added and the sentence was rephased as follows: ‘In another study [135], the high concentration of amino acid such as valine and leucine detected in the susceptible group of groupers infected with vibriosis is related to the activation of anti-bacterial infection pathways that involves valine, leucine, and isoleucine metabolism.’ Please refer to the page 10, line 442-445.
Line 401- these metabolites – what are the metabolites?
Response 9: Metabolites such as ‘valine and leucine’. Please refer to the page 10, line 445.
Line 412: significantly different? Are they increased or decreased? What are the implications of these changes on the health and survival of fish?
Response 10: The sentence was explained in detail as follows; Furthermore, a study conducted by Lardon et al. [136] on common carps (C. carpio) found that liver contained metabolites associated with the general energy pathway such as ATP/ADP, phosphocreatine, lactate, and amino acids metabolites such as valine, (iso)leucine, alanine, glutamate, glutamine, glycine, and aspartate were significantly decreased in fish that were induced with hypoxia compared to control fish. The decreased of these metabolites was due to oxygen limitation availability, which minimise the consumption of ATP/ADP and simultaneously decreased the accumulation of lactate to conserve their energy consumption [137]. As for amino acids, the decreased of the metabolites was due to the use of amino acid in regulating the growth and metabolism of the fish [138]. Please refer to the manuscript page 10, line 453-462.
Line 416: in regulation or to regulate – correct the typo
Response 11: Typo was corrected. Changed regulating to ‘to regulate’ in the sentence. Please refer page 11, line 509.
Lines 418-421: these sentences do not correlate and do not provide any meaningful information- authors should correct this.
Response 12: Rephase the sentence for better understanding of the statement. ‘In another study, the effects of waterborne chlorpyrifos in freshwater carp, Cyprinus carpio had cause the increased of biosynthesis of valine, leucine and isoleucine for oxidative/ stress metabolism, while the increased of, alanine and lactate pattern suggests that the subsequent metabolic pathway was inhibited.’ Please refer to the manuscript page 11, line 512-515.
Line 501- also identified 11 metabolites – correct this
Response 13: Mistake was corrected, we change ‘screened’ to ‘identified’ in the sentence. Please refer to page 17, line 601.

Reviewer 3 Report
The study “biology 1409515” is a review on the last aquaculture advancements about the management of infectious fish disease, the review is based on a summary of the main diseases in aquaculture and the possible developments, however, although the initial idea could be interesting, the paper should focus only on some aspects or be much more detailed and deal with all the topics in more depth.
For example: in line 92-The causative pathogens of vibriosis -how is it possible that the authors have forgotten to mention Vibrio anguillarum ?? a pathogen that causes massive losses to aquaculture and mariculture every year? (The authors must cite the major papers about vibriosis and V. anguillarum in particular such as: Crosa, Toranzo, Romalde etc..). I think that this part is too much superficial. Why do the authors mention only these two bacteria when there are many more bacteria that cause serious diseases in aquaculture? Because for example Photobacterium damselae subsp.piscida is not mentioned?
Therefore, although, the study will be of interest to BIOLOGY readers, I do not feel like recommending the work for publication. I recommend rewriting it by enlarging the missing parts or giving new life and a cut by highlighting only some aspects and then sending it back to the journal.
Moreover, I recommend to let a mother tongue see the text because there are many errors and sometimes the use of the chosen words is not scientific. For the moment I recommend rejection.
Reviewer 4 Report
The paper discusses the disease status, the immune status of aquatic organisms, multi-omics study status of study feasibility of joint application is reviewed, the paper for the future study of aquatic animal disease prevention and control to provide the direction. The paper detailed content, from the transcriptome study, proteomics, metabolomics etc various angles and examples are reviewed. There is a problem in this paper. Although multi-omics combined analysis can effectively solve some disease problems from the perspective of perspective, how can disease prevention and control be realized quickly and effectively with a macro high-throughput data set from DNA level to RNA level, as well as protein and metabolomics level? Standardized high-throughput omics analysis techniques and standardized experimental procedures are also needed to achieve the stability of research results.
Recommended publication.
Round 2
Reviewer 1 Report
The authors revised the manuscript but fall in a mistake. Please correct the details of the subheading "Active and passive immunization". Vaccination is active immunization, as it triggers cellular immunity to produce antibodies.
Have a look on this leading article to show the meaning of passive immunization in fish (https://www.jimmunol.org/content/198/11/4195).
Author Response
The authors revised the manuscript but fall in a mistake. Please correct the details of the subheading "Active and passive immunization". Vaccination is active immunization, as it triggers cellular immunity to produce antibodies.
Have a look on this leading article to show the meaning of passive immunization in fish (https://www.jimmunol.org/content/198/11/4195).
Response 1: Thank you for your comment. We have corrected the statement and added new citation. ‘There are two types of immunity known as passive and active immunity. Passive immunity can be defined as extraneous antibodies that temporary induce an immune response against pathogen [66]. Meanwhile, vaccination is an active immunity, is one of the most prominent approaches that have been used for strengthening host immunity and defending fish from both existing and future infections by triggering their cellular immunity to produce antibodies.’ Please refer to the manuscript in page 5, line 209-214.
Reviewer 3 Report
After re-reading the manuscript I noticed that the authors modified the text as suggested improving the manuscript, so I recommend the publication.
Author Response
After re-reading the manuscript I noticed that the authors modified the text as suggested improving the manuscript, so I recommend the publication.
Response 1: Thank you for your constructive review.
Round 3
Reviewer 1 Report
The authors revised the manuscript.